

# Cascade processes in rapidly rotating turbulence

Maxim Reshetnyak[1,2] and Oleg Pokhotelov[1]

[1]Schmidt Institute of Physics of the Earth of the Russian Academy of Sciences, B.Gruzinskaya, 10-1, Moscow, 123242, Russia
[2]Pushkov Institute of Terrestrial Magnetism, Ionosphere and Radio Wave Propagation RAS, Kalujskoe Hwy 4, Moscow, Troitsk, 108840, Russia

**Correspondence:** M. Reshetnyak (m.reshetnyak@gmail.com)

**Abstract.** The process of the kinetic energy and kinetic helicity transfer over the spectrum in an incompressible, rapidly rotating turbulent flow is considered. An analogue of the Fjortoft theorem for 3D rapidly rotating turbulence is proposed. It is shown that, similar to 2D turbulence, there are two cascades simultaneously: the inverse cascade of the kinetic energy and the direct cascade of the kinetic helicity, which in the case of 2D turbulence corresponds to the cascade of enstrophy. The proposed

scenario is in agreement with our earlier calculations, some recent numerical simulations, and physical experiment on rotating turbulence.

## 1  Introduction

The study of the spectral properties of the hydrodynamic turbulence includes both an analysis of the kinetic energy spectra itself and the physical fields that cause these motions, as well as the cascade processes, i.e. transfer of certain physical quantities

in the wave space. The latter, as a rule, reduces to study of the triad mechanisms (Frish, 1995; Verma, 2004), arising due to interactions of the Fourier harmonics in terms, containing products of the physical fields. In particular, for the Navier-Stokes equation, this is a non-linear term that ensures transfer of the quantities, that depend on the velocity field, over the spectrum. However the choice of these quantities is quite arbitrary, usually, transfer of the integrals of motions over the spectrum is considered. In its turn, the particular form of the integrals of motion depend on the dimensionality of the physical space, e.g.

(Lesieur, 2008). For 2D case they are the kinetic energy and enstrophy, while for 3D they are the kinetic energy and helicity.

The integral of motions are of the great importance because they determine the direction of transfer over the spectrum, caused by the triad mechanism. So, it is known that for 3D energy is transferred from the large scales, where the external force is applied, to the small scales, where dissipation is essential. The intermediate range of scales, where, in the absence of the external forces, there is only energy transfer across the spectrum, is called the inertial interval. This scenario of the energy

transfer corresponds to the so-called Kolmogorov's turbulence, according to which two large vortices interacting, produce a slightly smaller vortex. Since the scales of the vortices are close, it is customary to speak of a local energy transfer across the spectrum.

For 2D turbulence situation is reversed: the kinetic energy is transferred from the scale of the external force to a larger scale (the inverse energy cascade), where there is a sink of energy that depends on the model, e.g., the Rayleigh friction, generation





of the large-scale magnetic field in a dynamo and etc. The presence of the inverse cascade of the energy is well known in the atmospheric physics, where 2D approximation due to the thinness of the atmosphere along the vertical is frequently used (Pedlosky, 2013).

Historically, there is an important difference in study of 2D and 3D turbulences. For 2D, the presence of the inverse energy

cascade is a consequence of the energy conservation law and enstrophy (Fiorthof's theorem), e.g., (Fjørtoft, 1953), see also (Lesieur, 2008; Ditlevsen, 2010) and discussions of various applications in (Rose and Sulem, 1978; Kraichnan and Montgomery, 1980; Tabeling, 2002). On the contrary, the studies of 3D turbulence were initially carried out without discussing of kinetic helicity. It was silently assumed that in the isotropy approximation, used by default, the mean helicity, associated with the break of the mirror symmetry, is zero. In this case, for the non-rotating turbulence physical and numerical experiments indicated

presence of the direct energy cascade. These results were assumed to be the general rule for 3D turbulence. The justification either rebuttal of this statement was impossible, because the single conservation law for kinetic energy was used.

However, in the case of the geophysical turbulence, and especially its extreme form in the liquid cores of the planets, where the Rossby numbers are much less than unity, the net kinetic helicity is already non-zero. There, in the presence of the integral of motions for 3D convection, dependence of the fields on the coordinate along the axis of rotation degenerates: the velocity

field remains three-dimensional, but the velocity gradients along perpendicular coordinates, with respect to the rotation axis, are much greater than along the axis. It turns out that such an intermediate state between 2D and 3D inherits properties of the both two-dimensional and three-dimensional turbulence. This statement has the direct confirmations. Thus, the kinetic energy fluxes in the wave space demonstrate presence of two fluxes simultaneously: the both direct and inverse cascades, see the results of calculations of geostrophic turbulence in the Boussinesq approximation in (Reshetnyak and Hejda, 2008;

Hejda and Reshetnyak, 2009) in the plane layer, and in the spherical shell (Reshetnyak, M.Yu., Hejda, P., 2012), and references their in. The physical experiment with prescribed force also confirms existence of the inverse energy cascade (Campagne et al., 2014). Since the regimes considered above correspond to the case with a rapid rotation with a net kinetic helicity, it is reasonable to relate the existence of the inverse cascade of the kinetic energy to the presence of a second integral of motion, the conservation of the kinetic helicity. In what follows, for the fast rotating turbulence, where velocity and vorticity fields are

strongly correlated, we demonstrate existence of two cascades: an inverse cascade of the kinetic energy and the direct cascade of the kinetic helicity. This approach allows to explain the earlier results of the numerical simulations (Reshetnyak and Hejda, 2008; Hejda and Reshetnyak, 2009), and can be useful to analysis of the structure of triads in the wave space.

## 2   CONSERVATION LAWS AND CASCADES

Let us recall how the direction of the energy and enstrophy cascades in 2D turbulence is predicted, see for details (Lesieur,

2008,  p. 316), and (Ditlevsen, 2010, pp. 31). It is assumed that the turbulence is homogeneous and isotropic, so that all the fields depend on one scalar wave number $k$. The second assumption is that only three Fourier modes with the wave numbers





$k_1 \leq k_2 \leq k_3$ interact. Then the conservation law of the kinetic energy $E = V^2/2$ takes the form:

$$\delta E_1 + \delta E_2 + \delta E_3 = 0, \tag{1}$$

where $\delta E_i$ stands for the energy fluctuations with $i = 1 \ldots 3$.

Conservation of enstrophy $\Omega = \omega^2$, where $\boldsymbol{\omega} = \nabla \times \boldsymbol{V}$ is a vorticity, leads to the following equation for fluctuations $\delta \Omega_i = k_i^2 \delta E_i$:

$$\delta \Omega_1 + \delta \Omega_2 + \delta \Omega_3 = 0. \tag{2}$$

For the particular wave numbers $k_2 = 2k_1$ and $k_3 = 3k_1$, fluctuations $\delta E$, and $\delta \Omega$ are related as:

$$\delta E_1 = -\frac{5}{8}\delta E_2, \qquad \delta E_3 = -\frac{3}{8}\delta E_2,$$

$$\delta \Omega_1 = -\frac{5}{32}\delta \Omega_2, \qquad \delta \Omega_3 = -\frac{27}{32}\delta \Omega_2. \tag{3}$$

Let the mode with $k_2$ gives away energy $\delta E_2 < 0$, and find how it is redistributed among the modes with $k_1$ and $k_3$. Then $\delta \Omega_2 < 0$[1] and $\delta E_i$, $\delta \Omega_i$ for $i = 1$, and $i = 3$ are positive. The ratio $\delta E_1/\delta E_3 = 5/3 > 1$ corresponds to the inverse cascade of the kinetic energy, and $\delta \Omega_1/\delta \Omega_3 = 5/27 < 1$ to the direct cascade of the enstrophy. Below we consider how this approach, called the Fjortoft theorem, can be adopted to 3D geostrophic turbulence, where the conservation law for the enstrophy is replaced by the similar equation for the kinetic helicity.

The conservation of the kinetic helicity $H = \boldsymbol{V} \cdot \boldsymbol{\omega}$ for the triad interaction has the form:

$$\delta H_1 + \delta H_2 + \delta H_3 = 0, \tag{4}$$

with $\delta H$ for the helicity fluctuation.

The pseudoscalar kinetic helicity is the more complex quantity than the energy and enstrophy. Firstly, helicity in the general case, can change the sign. But this complexity is not crucial, since it is possible to consider domains, where $H$ possesses a fixed sign. We assume that the sign of $H$ coincides with the sign of the net helicity over the specific volume, e.g., the certain hemisphere for the convection in a spherical shell, or the half-volume in the plane layer with rigid horizontal boundaries. The accuracy of this approximation, which corresponds to the strongly correlated velocity and vorticity fields, is controlled in the models in the different ways. For the convection problem with the prescribed force it is determined by the specific form of the force. Fortunately, for this aim, it is more natural to use the self-consistent solution to the problem of the thermal convection in the geostrophic approximation, for which the degree of correlation $\boldsymbol{V}$ and $\boldsymbol{\omega}$ is extremely high (Hejda and Reshetnyak, 2010). In this case $H/E \sim 1/l$, where $l$ stands for the velocity scale, or $H \sim Ek$, where $k$ is the wave number.

The situation is more complicated with the fundamental possibility of the isotropy approximation application to the rapidly rotating turbulence, where the only one scalar $k$ is used. This can be commented as follows: the isotropy approximation for

---

[1]In more detail with arguments why the negative value of $\delta E_2$ is chosen is considered in (Lesieur, 2008). Briefly, it is concerned with the irreversibility of the diffusion process of the wave packet, initially localized at $k_2$ in the case of the free decaying turbulence.





**Table 1.** Exchange of the energy, $\delta E$, and kinetic helicity, $\delta H$, between the modes $k_1$, $k_2$, $k_3$ in units of $\delta E_2$; $\mathcal{I}$ – inverse, $\mathcal{D}$ – direct cascades.

| Quantity | $\delta E$ | | | | $\delta H$ | | | |
|---|---|---|---|---|---|---|---|---|
| Mode/Cascade | $k_1$ | $k_2$ | $k_3$ | Cascade | $k_1$ | $k_2$ | $k_3$ | Cascade |
| $A$ | $-\frac{1}{2}$ | $1$ | $-\frac{1}{2}$ | No | $-\frac{1}{2}$ | $2$ | $-\frac{3}{2}$ | $\mathcal{D}$ |
| $B$ | $-\frac{1}{4}$ | $1$ | $-\frac{3}{4}$ | $\mathcal{D}$ | $\frac{1}{4}$ | $2$ | $-\frac{9}{4}$ | $\mathcal{D}$ |
| $C$ | $-\frac{5}{4}$ | $1$ | $\frac{1}{4}$ | $\mathcal{I}$ | $-\frac{5}{4}$ | $2$ | $-\frac{3}{4}$ | $\mathcal{I}$ |
| $D$ | $-\frac{5}{2}$ | $1$ | $\frac{3}{2}$ | $\mathcal{I}$ | $\frac{5}{2}$ | $2$ | $-\frac{9}{2}$ | $\mathcal{D}$ |
| Sum over $A$, $B$, $C$, $D$ | $-\frac{9}{2}$ | $4$ | $\frac{1}{2}$ | $\mathcal{I}$ | $1$ | $8$ | $-9$ | $\mathcal{D}$ |

the case of the rapid rotation in some cases is still very successful (Zhou, 1995), when we are interesting in the estimates of the integral spectrum of the kinetic energy. Moreover, there are indications that products of the velocity and vorticity in the rapidly rotating flow for the different spatial directions have similar amplitudes and spatial profiles (Reshetnyak, 2017), which indirectly speaks in favor of the scalar $k$. Further, we also use the isotropy approximation, assuming that the vector fields

depend on one scalar $k$.

Now rewrite (4), having in mind the particular form $\delta H$, and that $k$ can change the sign. The latter leads to four cases $A$, $B$, $C$, and $D$, which degenerated in 2D due to the quadratic dependence of enstrophy on $k$:

$$p_1 \delta E_1 k_1 + \delta E_2 k_2 + p_2 \delta E_3 k_3 = 0, \tag{5}$$

where pairs of constants $(p_1, p_2)$ take the values: (A) – $(+1, +1)$, (B) – $(-1, +1)$, (C) – $(1, -1)$, (D) – $(-1, -1)$.

Solving the system (1) and (5) with respect to $\delta E_1$, and $\delta E_3$ with $k_2 = 2k_1$, $k_3 = 3k_1$, and having in mind relation between the energy and helicity, results in $\delta \boldsymbol{E} = \left(\delta E_1, \delta E_2, \delta E_3\right)$, and $\delta \boldsymbol{H} = \left(\delta H_1, \delta H_2, \delta H_3\right)$:

$$\delta \boldsymbol{E} = \left(\frac{3p_2 - 2}{p_1 - 3p_2}, 1, \frac{2 - p_1}{p_1 - 3p_2}\right) \delta E_2,$$

$$\delta \boldsymbol{H} = \left(p_1 \frac{3p_2 - 2}{p_1 - 3p_2}, 2, 3p_2 \frac{2 - p_1}{p_1 - 3p_2}\right) \delta E_2. \tag{6}$$

The values of the variations for the cases $(A - D)$ are given in Table 1. Taking into account that $\delta E_2 < 0$, we consider direction of the cascades in more details.

For $A$, there is no cascade, since the energy from $k_2$ is equally distributed between $k_1$, and $k_3$. The helicity cascade is direct, since $k_3$ has got helicity three times larger than $k_1$.

For $B$, the cascade of energy is direct, since the mode $k_3$ received three times more energy than $k_1$. For helicity, the cascade is also direct: the $k_1$ and $k_2$ modes transferred the helicity to the $k_3$ mode.

For $C$, the $k_2$ and $k_3$ modes transmit the energy to the $k_1$ mode, the inverse cascade of energy is accompanied by the inverse

helicity cascade. For $D$, inverse and direct cascades of energy and helicity are observed, respectively.





Since $\delta E_2$ is the same in $A - D$, we find the total variations, assuming that all the cases have the same probability, see Table 1. For the energy, the total cascade turns out to be inverse, and for helicity it is direct. The situation resembles the case for 2D, but instead of the inverse enstrophy cascade there is a helicity cascade.

To estimate how far the results depend on the specific choice of the values of $k_i$ in triads, we rewrite (5) as follows:

$p_1 \alpha \delta E_1 + \beta \delta E_2 + p_2 \delta E_3 = 0,$                                                     (7)

where real $\alpha$, and $\beta$ satisfy to the conditions: $0 \leq \beta \leq 1$, $0 \leq \alpha \leq \beta$. This approximation is valid for the large wave numbers with $\alpha \sim k_1/k_3$, $\beta \sim k_2/k_3$, where the discreteness of the wave numbers becomes insignificant. The joint solution of (1) and (7) has the form:

$$\delta \boldsymbol{E} = \left( \frac{\beta - p_2}{p_2 - \alpha p_1}, 1, \frac{\alpha p_1 - \beta}{p_2 - \alpha p_1} \right) \delta E_2,$$

$$\delta \boldsymbol{H} = \left( p_1 \alpha \frac{\beta - p_2}{p_2 - \alpha p_1}, \beta, p_2 \frac{\alpha p_1 - \beta}{p_2 - \alpha p_1} \right) \delta E_2.$$

                  (8)

Expressions (8) describe the energy and helicity fluctuations for arbitrary relations of the triangle' sides $(\alpha, \beta, 1)$.

Note that according to (8), the non-local cascade of the kinetic energy, predicted in the more sophisticated models (Waleffe, 1992), can exist. Let $\alpha \to 0$, $\beta \to 1$, then $\delta \boldsymbol{E} = \left( \frac{1 - p_2}{p_2}, 1, -\frac{1}{p_2} \right) \delta E_2$, and for $p_2 = -1$ the non-local inverse cascade of the kinetic energy takes place, where modes $k_2$, $k_3$ feed mode $k_1$.

Recalling that cases $A - D$ are equiprobable, we summarize each of the components of the vectors $\delta \boldsymbol{E}$ and $\delta \boldsymbol{H}$ in $p_1$ and 15   $p_2$, and divide it by 4, the number of cases:

$$\delta \boldsymbol{E}^{\Sigma} = \left( -\frac{1}{1 - \alpha^2}, 1, \frac{\alpha^2}{1 - \alpha^2} \right) \delta E_2,$$

$$\delta \boldsymbol{H}^{\Sigma} = \beta \left( \frac{\alpha^2}{1 - \alpha^2}, 1, -\frac{1}{1 - \alpha^2} \right) \delta E_2.$$

                  (9)

Note that, as $0 < \alpha < 1$, and $\delta E_2 < 0$, the components of the vector $\delta E_2^{\Sigma} = \delta E_2 < 0$, $\delta E_3^{\Sigma} < 0$, and $\delta E_1^{\Sigma} = \frac{\alpha^2}{1 - \alpha^2} \delta E_2 = -(\delta E_2^{\Sigma} + \delta E_3^{\Sigma}) > 0$. This corresponds to the inverse cascade of the energy. For $\alpha \to 0$ the total cascade from $k_3$ is zero. So far $0 < \alpha < \beta < 1$, the non-local cascade between $k_1$ and $k_2$ exists.

For the kinetic helicity situation is quite opposite: $\delta H_1^{\Sigma} = \frac{\alpha^2 \beta}{1 - \alpha^2} \delta E_2 < 0$, $\delta H_2^{\Sigma} = \beta \delta E_2 < 0$, and $\delta H_3^{\Sigma} = -\frac{\beta}{1 - \alpha^2} \delta E_2 = -(\delta H_1^{\Sigma} + \delta H_2^{\Sigma}) > 0$, the direct cascade of helicity takes place.

It is instructive to examine the case with exchange between $k_1$ and $k_3$, and $k_2$ for a catalyst. For energy this corresponds to the case when $\alpha \to 1$. In this case, the mode with $k_2$ does not participate in the energy exchange, and all energy from $k_3$ goes to $k_1$. Since $\beta > \alpha$, so that the both $\alpha \to 1$ and $\beta \to 1$, this corresponds to an equilateral triangle in the wave space, and to the 25   local transfer, correspondingly.

For helicity, a similar analysis yields: for $\alpha \to 1$, the $k_2$ mode participates in the exchange, and for $\alpha \to 0$ mode $k_1$ does not involve in the exchange.



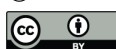

Expressions (9) describe the energy and helicity fluctuations for arbitrary relations of the sides of the triangle $(\alpha, \beta, 1)$. To find the total contribution for various $\alpha$, and $\beta$ one has to integrate over $\alpha$, and $\beta$. Let introduce the functional $\mathcal{I}(f) = \int_0^1 \int_0^\beta f(\alpha, \beta) \, d\alpha d\beta$, and calculate it for fluctuations $(1-\alpha^2)\delta E_i^\Sigma$, $(1-\alpha^2)\delta H_i^\Sigma$, substituting them instead of $f$ in $\mathcal{I}$, and denoting the result as $\delta \boldsymbol{E}^\int$, $\delta \boldsymbol{H}^\int$, respectively:

$$\delta \boldsymbol{E}^\int = \left(-\frac{6}{12}, \frac{5}{12}, \frac{1}{12}\right) \delta E_2,$$

$$\delta \boldsymbol{H}^\int = \left(\frac{1}{15}, \frac{4}{15}, -\frac{5}{15}\right) \delta E_2. \tag{10}$$

Relations (10) give an idea of the total ratio of the contributions of modes $k_i$ to the inverse cascade of kinetic energy and to the direct cascade of the kinetic helicity over the spectrum. As it was predicted by estimates with the particular choice of the wave numbers above, there are two cascades of the kinetic energy and helicity, one of that is direct, and the second one is inverse. For the both cascades all three modes participate in the exchange. Contribution of the intermediate mode $k_2$ is comparable to the both other modes.

## 3  Conclusions

Inspection of the recent papers (Mininni and Pouquet, 2010; Biferale et al., 2017) on cascades in 3D rotating turbulence reveals that inverse cascade of the kinetic energy is quite common phenomenon, however its details depend on the parameter regimes, such as, e.g., the angular rotation velocity (Mininni et al., 2009; Campagne et al., 2014), scale of the prescribed force (Deusebio et al., 2014). In this sense the considered above simple scenario of the inverse cascade for the energy reflects general properties of the rotating turbulence, and seems to be quite natural. Dependence on parameters make study more complicated and one way to overcome this difficulty is to consider more realistic forms of the prescribed force. By this we mean that the force should be tested for consistency with the self-consistent natural convection, e.g. thermal convection in presence of the rapid rotation. Such test will exclude possibility when the increase of the angular rotation velocity will cause transition from 3D to 2D. This situation with unclear transition from one set of integral of motions to the other one, can be unreal in principle. Thus, for the thermal convection (the same for compositional one) increase of rotation suppresses convection. But if thermal convection has already started in the system with collinear gravity and axis of rotation, then the flow is geostrophic but still three-dimensional with similar in amplitudes of velocity components, see (Roberts, 1965). Consequently, the models, which demonstrate something different, usually omit buoyancy at all, either do not take into account dependence of the critical Rayleigh number on rotation. The back side of the coin is that fully self-consistent models, as a rule, have worse spatial resolution, and may be what is more important, in such models, due to absence of the sink of the energy at large scales, the statistical equilibrium there is developed (Ditlevsen, 2010), decreasing amplitude of the inverse flux, and making interval of the spectra, where the inverse cascade takes place, even shorter. The obtained results can be interesting for testing of the turbulent diffusion coefficients in geostrophic flows, see for details (Smith et al., 2002).





*Acknowledgements.* MR acknowledges financial support from the Russian Science Foundation (grant N°16-17-10097).

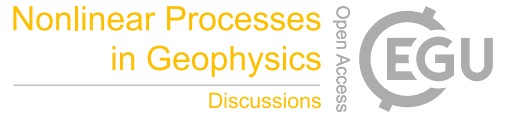

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
