# Peer review of "Cascade processes in rapidly rotating turbulence"

_Nonlinear Processes in Geophysics, 2018_

## Referee Comment (RC1) · Anonymous Referee #1 · 24 Sep 2018

General comments: The authors present a theoretical study of hydrodynamical (HD) fluid turbulence under fast rotation. The discussion is focussed on the cascade processes (direct versus inverse) by using an analogue of the Fjortoft theorem. They conclude that two cascades of different directions are simultaneously present. This new result is in agreement with previous calculations made by the authors. The subject, rotating turbulence, is a widely studied subject. Several experiments have been designed to study the cascade processes, and 3D direct numerical simulations as well as analytical treatments have also been done. Here, it is proposed to use similar arguments as for 2D HD turbulence; in particular the assumption of isotropy is used. This assumption is, however, in contradiction with basic properties of rapidly rotating turbulence where strong anisotropy is observed and predicted. This property can change

drastically the conclusion. Therefore, I do not think that this paper, as it is currently proposed, gives significant results for this problem. In conclusion, I do not recommend this paper for a publication in NPG.

Specific comments: The anisotropic character of rotating turbulence is well recognized for rotating turbulence. For example, in the limit of fast rotation (which is the subject of this paper) a weak turbulence description is possible for which a direct cascade of energy and helicity is found analytically (Galtier, PRE, 2003, 2014). This behavior was already predicted by Waleffe (PoF, 1993) and studied by Cambon et al. (JFM, 1989; 1997). All these theoretical papers are ignored by the authors whereas there are fundamental for this problem. Full credit should also be given to (recent) experiments (see e.g. the review by Godeferd & Moisy, AMR, 2015). A nonlocal inverse cascade of energy is, however, possible but it may imply a non-trivial relation with the slow mode (k//=0), which cannot be described in the present paper. This point is also discussed in the recent paper by Buzzicotti et al. (PRF, 2018). I think this contribution could be very interesting if the authors can generalize their formalism to non-isotropic turbulence.

Technical corrections: Frish, 1995 -> Frisch, 1995

---

## Referee Comment (RC2) · Anonymous Referee #2 · 1 Oct 2018

the paper is about the existence of an inverse cascade in rotating flows. Using a systematic analysis about the energy enstrophy and helicity evolution on a restricted isolated set of triads they conclude about the potential existence of an inverse energy cascade and a direct helicity cascade in the presence of a strong correlation among vorticity and velocity a limit also known as 2d3c dynamics. On one side, the problem is potentially of interest for many geophysical and theoretical applications. On the other side the paper has many limitations and too many unclear steps/assumptions. The result is that it cannot be published in this form. I list my main criticisms:

1) it is known that rotating turbulence undergoes a transition from direct to split-cascade (both direct and inverse energy fluxes) at increasing Omega. There are no hints of a purely inverse cascade unless the system is purely 2d, the forcing acts only on the slow

modes, or the system is purely homochiral (forced to evolve on same-sign of helicity as done by biferale, toschi and musacchio et al PRL 2012). This fact is completely absent in the discussion and they do not even have Omega (rotation rate) as a control parameter.

2) It is argued that in the infinite volume limit the inverse cascade should disappear (at least this is wave wave-turbulence predicts). No discussion about this effect is present in their calculations

3) There exists many analytical and numerical studies of helicity effects in rotating ad non-rotating turbulence (see smith and waleffe POF 1999, chen et al JFM 2005, biferale et al PRX 2016, alexandros alexakis JFM 2017) which are completely ignored. It is not clear why they do not distinguish homo and heterochiral triads in their discussion.

4) Even more importantly, it is not true that there exists coherent regions with same-sign of helicity in turbulent rotating flows, even if mean helicity is non zero, it is always the results of non-trivial and non-exact small-scale cancellations of regions with different helicity signs.

5) Equations of motions and the exact approximation they are imposing must be cited explicitly. It is absolutely non clear what is the system they are considering. They need to start from the rotating case, saying explicitly the boundary conditions (see also point 1), how do they force the system, what are the approximation made to reduce to the 3 shells limit in their paper etc.

6) The whole discussion is based on isotropic statistics which is at odd with the 2d3c limit.

In conclusions, the analytical set up is not clear enough to control the set of approximation made and to assess the potential limitations.